# Reproducibility study of FairAC

**Gijs de Jong**[*]                                          *gijs.de.jong@student.uva.nl*
*University of Amsterdam*

**Macha J. Meijer**[*]                                       *macha.meijer@student.uva.nl*
*University of Amsterdam*

**Derck W. E. Prinzhorn**[*]                                 *derck.prinzhorn@student.uva.nl*
*University of Amsterdam*

**Harold Ruiter**[*]                                         *harold.ruiter@student.uva.nl*
*University of Amsterdam*

**Reviewed on OpenReview:** `https://openreview.net/forum?id=ccDi5jtSF7`

## Abstract

This work aims to reproduce the findings of the paper "Fair Attribute Completion on Graph with Missing Attributes" written by Guo, Chu, and Li [1] by investigating the claims made in the paper. This paper suggests that the results of the original paper are reproducible and thus, the claims hold. However, the claim that FairAC is a generic framework for many downstream tasks is very broad and could therefore only be partially tested. Moreover, we show that FairAC is generalizable to various datasets and sensitive attributes and show evidence that the improvement in group fairness of the FairAC framework does not come at the expense of individual fairness. Lastly, the codebase of FairAC has been refactored and is now easily applicable for various datasets and models.

## 1 Introduction

In recent years, graphs are an increasingly common data structure used in real-world applications [2]. Graph neural networks (GNNs) have shown promising performance in a large range of tasks, such as community detection [3], node classification [4], link prediction [5] and graph classification [6]. However, there is a possibility that the graph data may be biased, for example due to under-representation of a specific group [7]. If the model is trained on this biased graph, the model may be unfair with respect to certain sensitive attributes such as demographic or gender [8, 9]. Aside from this, the node information of graph datasets often is incomplete, for example when a node was recently added to a network [10].

To address the problem of attribute completion and fairness in graph data, the FairAC framework was proposed by Guo, Chu, and Li [1]. FairAC is a fair attribute completion framework which can be applied to a dataset, after which the result can be used as input for many downstream tasks, such as node classification or link prediction.

The aim of this paper is to reproduce the original paper of FairAC and extend on the original paper. In summary, the following contributions are made:

1. The original FairAC paper is reproduced and the original claims are analysed.

2. The codebase of FairAC is improved and more easily applicable to other datasets and models.

---

[*]Equal contribution.

3. The original paper is analyzed further by using new datasets, testing the framework on various sensitive attributes and evaluating the performance on individual fairness.

## 2 Scope of reproducibility

This study describes the reproducibility of 'Fair Attribute Completion on Graph with Missing Attributes' by Guo, Chu, and Li [1]. In earlier research, various fair graph methods have been developed, such as Fairwalk [11] and fair GraphSAGE [12]. While previous research addressed fairness in graphs with partially incomplete attributes [7], FairAC uniquely targets nodes with entirely missing attributes. This approach also diverges from methods like those in [7] by addressing both node feature unfairness and topological unfairness. In the context of fairness, feature unfairness refers to bias within a single node's embedding, which can be viewed as unfair when it contains sensitive information. Topological unfairness arises when aggregating neighbouring nodes introduces sensitive information in the final embedding [13]. The paper proposes FairAC, a fair attribute completion model for graphs. For more details on this method, see Section 3.

The claims that the paper made are as follows:

*Claim 1* A new framework, namely FairAC, is proposed, which can be used for fair graph attribute completion and addresses both feature and topological unfairness in the resulting graph embeddings.

*Claim 2* FairAC is generic and can be used in many graph-based downstream tasks.

*Claim 3* FairAC is effective in eliminating unfairness while maintaining an accuracy comparable to other methods.

*Claim 4* FairAC is effective even if a large amount of the attributes are missing.

*Claim 5* Adversarial learning is necessary to obtain a better performance on group fairness.

Beyond reproducing the original study, we extended our work to assess FairAC's generalizability. This involved evaluating the framework across different datasets and varying sensitive attributes. Additionally, we conducted a more in-depth analysis of FairAC's fairness capabilities, using a metric for individual fairness.

## 3 Methodology

The FairAC implementation is openly accessible, but the baseline code for GCN and FairGNN is not included in this repository. To address this, we integrated these separate codebases, which are also publicly available, into a unified framework. In the restructuring, we enhanced the codebase to support the use of various datasets and different sensitive attributes in combination with FairAC. Furthermore, we expanded FairAC's evaluation criteria by incorporating a measure of individual fairness, offering contrast to the original paper's focus on group fairness.

### 3.1 Model description

FairAC is novel framework designed to ensure fairness in attribute completion for graph nodes, as depicted in Figure 1. This framework is composed of roughly three components.

1. **Auto-encoder**. The auto-encoder is utilized to generate node embeddings of complete nodes. This involves adversarial training with a sensitive classifier tasked with detecting the presence of sensitive information in embeddings, guiding the auto-encoder towards more fair feature representation.

2. **Attention-based attribute completion**. FairAC employs an attention mechanism to determine the importance of neighbouring nodes when completing the target node's attributes. The final embedding of the target node is created through a weighted aggregation, with the weights assigned by the attention mechanism.

3. **Topological fairness**. The sensitive classifier reappears in this component to evaluate topological fairness of the target node's embedding, which is completely based on neighbouring nodes. If sensitive information is present, it indicates that it originated from the aggregation of neighboring nodes, highlighting potential topological bias.

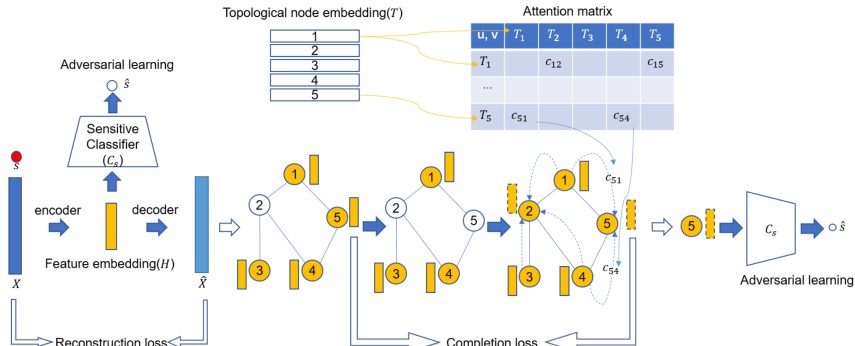

Figure 1: The FairAC framework consists of mainly three components: An auto-encoder to generate embeddings, an attention-based mechanism for attribute completion and a sensitive classifier to apply adversarial learning [1].

FairAC's loss function integrates these three components (detailed in Appendix A). A hyperparameter $\beta$ is used to adjust the influence of the sensitive classifier on the overall loss. $\beta$ is multiplied with the loss of the sensitive classifier in both the topological and the node embedding prediction parts. The loss function is defined as $\mathcal{L} = \mathcal{L}_F + \mathcal{L}_C + \beta\mathcal{L}_T$, where $\mathcal{L}_F$ represents the feature fairness component loss, defined as $\mathcal{L}_F = \mathcal{L}_{ae} - \beta\mathcal{L}_{C_s}$.

## 3.2 Datasets

The authors have made available `csv` files of the three real-world datasets used for their experiments. We follow the paper and reproduce their evaluations on multiple datasets for each method. We present the relevant datasets in detail in Table 5.

- **NBA dataset**. This is an extended version of a Kaggle dataset[1] containing performance statistics of approximately 400 NBA basketball players from the 2016-2017 season. It includes personal information such as age, nationality, gender and salary [7], with player interactions on Twitter defining the network relationships. The node label indicates whether a player's salary is above the median and nationality serves as the sensitive attribute, intended to be excluded from the embeddings.

- **Pokec datasets**. Derived from Pokec, a Slovakian social network [14], the Pokec-z and Pokec-n datasets are adapted from Dai and Wang [7]. While the original FairAC paper used region as the sensitive attribute for both Pokec datasets, our study also evaluates age and gender. Age is converted to a binary variable by indicating if a person is younger than 21 years or not, similar to the approach in the credit dataset [15].

Additional datasets, German Credit and Recidivism, were also used to evaluate FairAC [15]. These datasets were selected for their different themes and small size, making attribute completion particularly useful due to the higher importance of each data point. The code to preprocess the dataset was adapted from PyGDebias[2]. For FairAC training, the first 25% of each dataset was used, a limitation due to memory constraints of the machines used for the original paper. The subsequent GNN training utilized 50% of the data, with the remaining 25% reserved for evaluation. All data splits follow those used in the original paper.

---

[1]https://www.kaggle.com/datasets/noahgift/social-power-nba
[2]https://github.com/yushundong/PyGDebias/

### 3.3 Hyperparameters

In order to reproduce the experiments of the original author as closely as possible, the hyperparameters used in the original paper were used when available. This means that the feature drop rate ($\alpha$) is 0.3 unless mentioned otherwise. $\beta$ is 1.0 for all datasets, except for pokec-n, where $\beta$ is equal to 0.5. In case of missing hyperparameters, the parameters in the provided shell scripts were adapted. For training, 3000 epochs were used including 200 epochs used for autoencoder pretraining. An Adam optimizer was used with a learning rate of 0.001 and a weight decay of $1 \cdot 10^{-5}$. The details of all hyperparameters can be found in Appendix G.

In the original paper, all experiments are run three times. We assumed that this was done on three different seeds. Since, in the available scripts, only one seed is stated, we've decided to use the seeds 40, 41 and 42, per advice of the authors.

### 3.4 Evaluation metrics

Model performance is evaluated in terms of accuracy, AUC and fairness. When evaluating fairness, an important distiction can be made between two fairness types: group fairness and individual fairness. Group fairness measures equality between different protected groups, while individual fairness measures the similarity in classification outcome for similar individuals [16]. The original paper uses two important metrics to evaluate in terms of (group) fairness: statistical parity and equal opportunity. Additionally, we add an individual fairness metric, consistency. The label $y$ denotes the ground-truth node label and the sensitive attribute $s$ indicates a sensitive group. For example, we consider a binary node classification task and two sensitive groups $s \in \{0, 1\}$.

**Statistical Parity**. Statistical Parity Difference ($\Delta$SP) [9] is used to capture the difference in model prediction outcomes for different sensitive groups:

$$\Delta SP = P(\hat{y}|s = 0) - P(\hat{y}|s = 1) \tag{1}$$

**Equal Opportunity**. Equal Opportunity Difference ($\Delta$EO) [17] is used to capture the difference in true positive rates for different sensitive groups:

$$\Delta EO = P(\hat{y} = 1|s = 0, y = 1) - P(\hat{y} = 1|s = 1, y = 1) \tag{2}$$

**Consistency**. Consistency is used to capture the individual fairness by measuring the consistency of outcomes between individuals who are similar to each other [18]. The formal definition regarding a fairness similarity matrix $S^F$ is:

$$consistency = 1 - \frac{\sum_i \sum_j |y_i - \hat{y}_j| \cdot S_{ij}^F}{\sum_i \sum_j S_{ij}^F}, \quad \forall i \neq j. \tag{3}$$

### 3.5 Experimental setup and code

All specific experiments conducted are described below, together with a description of the training details and the code.

#### 3.5.1 Model training

The original paper provided a brief overview of the training algorithm; for more details refer to [1]. Our analysis found that certain critical details were missing in the original description. In the first part, the data split, it was found that the AC model is trained on only 25% of the nodes in the dataset, and during the test on the downstream task, the full 100% of the graph is used to complete any missing attributes and create new embeddings for the downstream task. After the data is split, the training process is started. An interesting detail in this process is that the auto-encoder in the AC model is pre-trained for 200 iterations. This iteration count remains fixed across all datasets, meaning that for smaller datasets the auto-encoder is pre-trained using comparatively less data, which is the case in for example the NBA or German credit

dataset, as described in Section 3.2. Furthermore, the auto-encoder and attribute completion model are optimized differently during pre-training than during the training iterations. During pre-training they are optimized using $\mathcal{L}_{ae} + \mathcal{L}_C$. This to be expected as the sensitive classifier is not being trained yet.

Evaluation of the model is done for the first time after 1000 epochs. Every 200 epochs, a GCN is trained on the embeddings generated by FairAC. This GCN is evaluated and provides results for the performance of FairAC. The final results reported are the results of the model with the best fairness performance that scores above the accuracy and AUC threshold that are set at the start of training. For all datasets, the thresholds were set to 0.65 and 0.69, respectively. The complete model training procedure for the FairAC model is given in Appendix C.

### 3.5.2 Code

The original paper published their code, including shell scripts to run specific experiments, based on the FairGNN codebase[3]. However, the code was difficult to work with and contained several peculiarities that are not mentioned in the original work (as mentioned in Section 3.5.1). In order to run the code on additional datasets and downstream tasks, the FairAC framework has been rewritten into an ergonomic library that can be used for any GNN without any significant changes to the code. The source code for this reproduction has been made available (See Appendix D for more details). In addition to creating the library, the implementation has been made faster, which makes the runtime about 33% faster.

### 3.5.3 Experiment reproducibility study

To accurately reproduce the original study, various experiments were conducted. Firstly, to verify claim 1, 2 and 3, the FairAC framework was applied on the default GCN on the datasets NBA, pokec-z and pokec-n. The results of this were compared with the results of using only a GCN and using the FairGNN method. Conducting experiments on different datasets provides evidence for claim 2 specifically. To verify claim 4, experiments on all three models and the FairAC model without adversarial training were conducted with different attribute missing rates, and the results are compared. Similarly, for claim 5, the FairAC model was trained with different $\beta$ values, to see the effect of adversarial learning.

### 3.5.4 Experiments beyond original paper

To further analyze the FairAC framework, additional experiments were conducted. To provide additional evidence for claim 2, the framework was tested on two extra datasets, as described in Section 3.2. Additionally, the framework was tested on different sensitive attributes, to provide further evidence for generalizability. Specifically, two additional experiments were run on the pokec-z dataset, with the sensitive attributes age and gender. Lastly, to test whether there is a trade-off between the group fairness provided by FairAC and individual fairness, all experiments were tested on an additional metric, consistency, which measures the individual fairness. The consistency between the default GCN and FairAC was compared to see the influence of FairAC on individual fairness.

### 3.6 Computational requirements

All experiments were performed on 1 NVIDIA Titan RTX GPU. Training one FairAC model takes about 30 minutes on this GPU, while training a (Fair)GNN model takes about 10 minutes. Therefore, reproducing all original experiments costs about 31 GPU hours. Roughly 15 GPU hours were used for performing the additional experiments. Thus, 46 GPU hours were used in total. More details about the GPU usage can be found in Appendix F.

---

[3]https://github.com/EnyanDai/FairGNN

## 4 Results

In this section, the results of the reproducibility study are described. This includes all experiments ran to verify the claims as described in Section 2. In addition to this, the results of the work done beyond the original study are described.

### 4.1 Results reproducibility study

In Table 1, the reproduced results of the original study are shown. Together with this, the performance on individual fairness is shown, which are analyzed in Section 4.2. In the original study, the main trends observed are that fairAC performs similarly to other methods on accuracy and AUC and outperforms all other alternatives on statistical parity and equal opportunity. The same trends can be observed in the results from the reproducibility study. The accuracy and AUC scores are similar, although not as good as FairGNN. On statistical parity and equal opportunity, FairAC outperforms all other methods on almost every dataset. The actual numbers of all methods are slightly different from the original paper. This could be due to the optimization method not being deterministic. These results verify claim 1 and 3, as listed in Section 2. Namely, FairAC addresses unfairness better than other methods while it maintains a comparable accuracy.

| Dataset | Method | M | Acc ↑ | AUC ↑ | ΔSP ↓ | ΔEO ↓ | ΔSP+ΔEO ↓ | Consistency ↑ |
|---|---|---|---|---|---|---|---|---|
| NBA | GCN | ✓ | $66.98 \pm 1.18$ | $\mathbf{76.15 \pm 1.40}$ | $0.14 \pm 0.13$ | $0.57 \pm 0.06$ | $0.71 \pm 0.18$ | $\mathbf{2.64 \pm 0.00}$ |
| | ALFR | × | $64.3 \pm 1.3$ | $71.5 \pm 0.3$ | $2.3 \pm 0.9$ | $3.2 \pm 1.5$ | $5.5 \pm 2.4$ | - |
| | ALFR-e | × | $66.0 \pm 0.4$ | $72.9 \pm 1.0$ | $4.7 \pm 1.8$ | $4.7 \pm 1.7$ | $9.4 \pm 3.4$ | - |
| | Debias | × | $63.1 \pm 1.1$ | $71.3 \pm 0.7$ | $2.5 \pm 1.5$ | $3.1 \pm 1.9$ | $5.6 \pm 3.4$ | - |
| | Debias-e | × | $65.6 \pm 2.4$ | $72.9 \pm 1.2$ | $5.3 \pm 0.9$ | $3.1 \pm 1.3$ | $8.4 \pm 2.2$ | - |
| | FCGE | × | $66.0 \pm 1.5$ | $73.6 \pm 1.5$ | $2.9 \pm 1.0$ | $3.0 \pm 1.2$ | $5.9 \pm 2.2$ | - |
| | FairGNN | ✓ | $\mathbf{68.39 \pm 3.12}$ | $74.29 \pm 1.19$ | $2.81 \pm 4.01$ | $3.00 \pm 4.07$ | $5.81 \pm 8.08$ | $2.64 \pm 0.00$ |
| | FairAC (Ours) | ✓ | $66.51 \pm 1.09$ | $75.69 \pm 1.31$ | $\mathbf{0.09 \pm 0.08}$ | $\mathbf{0.10 \pm 0.00}$ | $\mathbf{0.19 \pm 0.08}$ | $2.64 \pm 0.00$ |
| Pokec-z | GCN | ✓ | $65.10 \pm 0.24$ | $68.42 \pm 0.12$ | $1.72 \pm 1.17$ | $1.37 \pm 0.51$ | $3.08 \pm 1.68$ | $\mathbf{41.35 \pm 0.01}$ |
| | ALFR | × | $65.4 \pm 0.4$ | $71.3 \pm 0.3$ | $2.8 \pm 0.5$ | $1.1 \pm 0.4$ | $3.9 \pm 0.9$ | - |
| | ALFR-e | × | $68.0 \pm 0.6$ | $74.0 \pm 0.7$ | $5.8 \pm 0.4$ | $2.8 \pm 0.8$ | $8.6 \pm 1.2$ | - |
| | Debias | × | $65.2 \pm 0.7$ | $71.4 \pm 0.6$ | $1.9 \pm 0.6$ | $1.9 \pm 0.4$ | $3.8 \pm 1.0$ | - |
| | Debias-e | × | $67.5 \pm 0.7$ | $74.2 \pm 0.7$ | $4.7 \pm 1.0$ | $3.0 \pm 1.4$ | $7.7 \pm 2.4$ | - |
| | FCGE | × | $65.9 \pm 0.2$ | $71.0 \pm 0.2$ | $3.1 \pm 0.5$ | $1.7 \pm 0.6$ | $4.8 \pm 1.1$ | - |
| | FairGNN | ✓ | $\mathbf{68.16 \pm 0.59}$ | $\mathbf{75.67 \pm 0.52}$ | $1.56 \pm 0.45$ | $3.17 \pm 1.07$ | $4.73 \pm 1.47$ | $41.35 \pm 0.01$ |
| | FairAC (Ours) | ✓ | $65.33 \pm 0.30$ | $71.20 \pm 1.74$ | $\mathbf{0.55 \pm 0.10}$ | $\mathbf{0.13 \pm 0.15}$ | $\mathbf{0.68 \pm 0.09}$ | $41.33 \pm 0.00$ |
| Pokec-n | GCN | ✓ | $\mathbf{67.88 \pm 1.46}$ | $\mathbf{72.86 \pm 1.44}$ | $3.22 \pm 1.29$ | $5.93 \pm 2.76$ | $9.15 \pm 4.05$ | $45.93 \pm 0.00$ |
| | ALFR | × | $63.1 \pm 0.6$ | $67.7 \pm 0.5$ | $3.05 \pm 0.5$ | $3.9 \pm 0.6$ | $3.95 \pm 1.1$ | - |
| | ALFR-e | × | $66.2 \pm 0.4$ | $71.9 \pm 1.0$ | $4.1 \pm 1.8$ | $4.6 \pm 1.7$ | $8.7 \pm 3.5$ | - |
| | Debias | × | $62.6 \pm 1.1$ | $67.9 \pm 0.7$ | $2.4 \pm 1.5$ | $2.6 \pm 1.9$ | $5.0 \pm 3.4$ | - |
| | Debias-e | × | $65.6 \pm 2.4$ | $71.7 \pm 1.2$ | $3.6 \pm 0.9$ | $4.4 \pm 1.3$ | $8.0 \pm 2.2$ | - |
| | FCGE | × | $64.8 \pm 1.5$ | $69.5 \pm 1.5$ | $4.1 \pm 1.0$ | $5.5 \pm 1.2$ | $9.6 \pm 2.2$ | - |
| | FairGNN | ✓ | $67.06 \pm 0.37$ | $71.58 \pm 2.58$ | $0.55 \pm 0.50$ | $\mathbf{0.30 \pm 0.20}$ | $0.85 \pm 0.31$ | $45.93 \pm 0.00$ |
| | FairAC (Ours) | ✓ | $67.00 \pm 1.93$ | $72.57 \pm 1.68$ | $\mathbf{0.11 \pm 0.06}$ | $0.47 \pm 0.81$ | $\mathbf{0.58 \pm 0.76}$ | $\mathbf{45.94 \pm 0.02}$ |

Table 1: Comparison of FairAC with FairGNN on the nba, pokec-z and pokec-n dataset. The methods are applied on the GCN classifier. The best results are denoted in bold. The ALFR, ALFR-e, Debias, Debias-e and FCGE baselines are adapted from Guo, Chu, and Li [1].

In addition to the main comparison of different baselines, an ablation study on different attribute missing rates ($\alpha$) was done in the original paper to verify claim 4. The results are shown in Table 2. The trends in the reproduced results are very similar to the original trends, as FairAC performs best on fairness for different $\alpha$. However, for $\alpha = 0.1$, it is observed that in the reproduced results, BaseAC, which is the same architecture as FairAC but without adversarial training, performs slightly better than FairAC. In addition to this, it is observed that adversarial training is more useful with a large attribute missing rate, an observation that wasn't made in the original paper.

Although the original results are reproduced, the reproduction would not have been possible without access to the codebase. A lot of the training details, such as hyperparameters and pretraining of the auto-encoder were not mentioned in the paper, only in the codebase. In addition to this, the random seeds used in the original study were only retrieved after contact with the authors.

| $\alpha$ | Method | Acc ↑ | AUC ↑ | ΔSP ↓ | ΔEO ↓ | ΔSP+ΔEO ↓ | Consistency ↑ |
|---|---|---|---|---|---|---|---|
| 0.1 | GCN | $65.93 \pm 0.33$ | $69.86 \pm 1.73$ | $3.52 \pm 2.94$ | $2.73 \pm 3.18$ | $6.26 \pm 6.12$ | $41.33 \pm 0.00$ |
| | FairGNN | $\mathbf{68.67 \pm 2.52}$ | $\mathbf{76.95 \pm 0.53}$ | $1.97 \pm 0.52$ | $2.23 \pm 0.31$ | $4.20 \pm 0.59$ | $41.33 \pm 0.00$ |
| | BaseAC | $66.16 \pm 0.11$ | $69.22 \pm 0.07$ | $\mathbf{0.08 \pm 0.01}$ | $\mathbf{0.40 \pm 0.20}$ | $\mathbf{0.48 \pm 0.21}$ | $41.33 \pm 0.00$ |
| | FairAC | $66.04 \pm 0.69$ | $70.72 \pm 1.50$ | $0.26 \pm 0.28$ | $0.50 \pm 0.62$ | $0.76 \pm 0.90$ | $41.33 \pm 0.00$ |
| 0.3 | GCN | $65.10 \pm 0.24$ | $68.42 \pm 0.12$ | $1.72 \pm 1.17$ | $1.37 \pm 0.51$ | $3.08 \pm 1.68$ | $41.35 \pm 0.01$ |
| | FairGNN | $\mathbf{68.16 \pm 0.59}$ | $\mathbf{75.67 \pm 0.52}$ | $1.56 \pm 0.45$ | $3.17 \pm 1.07$ | $4.73 \pm 1.47$ | $41.35 \pm 0.01$ |
| | BaseAC | $66.26 \pm 0.32$ | $71.11 \pm 1.73$ | $\mathbf{0.35 \pm 0.15}$ | $0.73 \pm 1.18$ | $1.08 \pm 1.32$ | $41.33 \pm 0.00$ |
| | FairAC | $65.33 \pm 0.30$ | $71.20 \pm 1.74$ | $0.55 \pm 0.10$ | $\mathbf{0.13 \pm 0.15}$ | $\mathbf{0.68 \pm 0.09}$ | $41.33 \pm 0.00$ |
| 0.5 | GCN | $65.65 \pm 0.97$ | $69.72 \pm 2.15$ | $2.75 \pm 2.09$ | $3.50 \pm 3.32$ | $6.25 \pm 5.40$ | $41.38 \pm 0.02$ |
| | FairGNN | $\mathbf{65.97 \pm 0.56}$ | $\mathbf{72.99 \pm 0.44}$ | $2.40 \pm 1.44$ | $3.07 \pm 2.41$ | $5.47 \pm 3.84$ | $41.37 \pm 0.01$ |
| | BaseAC | $65.45 \pm 0.40$ | $70.64 \pm 1.10$ | $0.13 \pm 0.20$ | $0.33 \pm 0.58$ | $0.47 \pm 0.77$ | $41.33 \pm 0.00$ |
| | FairAC | $65.62 \pm 0.02$ | $71.11 \pm 1.02$ | $\mathbf{0.05 \pm 0.05}$ | $\mathbf{0.30 \pm 0.52}$ | $\mathbf{0.35 \pm 0.50}$ | $41.33 \pm 0.00$ |
| 0.8 | GCN | $65.37 \pm 1.30$ | $\mathbf{71.62 \pm 2.33}$ | $5.10 \pm 3.12$ | $5.53 \pm 3.86$ | $10.64 \pm 6.84$ | $41.47 \pm 0.04$ |
| | FairGNN | $63.81 \pm 0.50$ | $67.57 \pm 0.30$ | $2.96 \pm 0.28$ | $1.77 \pm 1.33$ | $4.73 \pm 1.10$ | $41.44 \pm 0.02$ |
| | BaseAC | $\mathbf{66.03 \pm 0.69}$ | $71.06 \pm 1.21$ | $0.32 \pm 0.39$ | $0.67 \pm 0.65$ | $0.99 \pm 0.86$ | $41.33 \pm 0.00$ |
| | FairAC | $65.38 \pm 0.14$ | $71.51 \pm 0.68$ | $\mathbf{0.23 \pm 0.37}$ | $\mathbf{0.03 \pm 0.06}$ | $\mathbf{0.27 \pm 0.43}$ | $41.33 \pm 0.00$ |

Table 2: Comparison of FairAC with FairGNN on the pokec-z dataset with different attribute missing rates ($\alpha$). The methods are applied on the GCN classifier. The best results are denoted in bold.

Furthermore, a hyperparameter study of $\beta$ was done in the original paper. $\beta$ balances fairness and accuracy (details in Section 3.1). The original study introduced adversarial training to improve group fairness and claimed that adversarial training is necessary for a good performance on group fairness, based on $\beta$ hyperparameter study. It reported that increasing $\beta$ improves fairness but slightly reduces accuracy by about 0.5%. However, the lack of standard deviation in the original figure makes it unclear whether the experiment was done three times or one. In this study, the experiment is done three times. The mean and standard deviation are shown in Figure 2. In general, the same trends are observed. Notably, at $\beta = 0.8$, we observed a marginal increase in fairness, contrary to the original study's continuous decline. From this, it can be concluded that adversarial learning helps fairness performance in the FairAC framework.

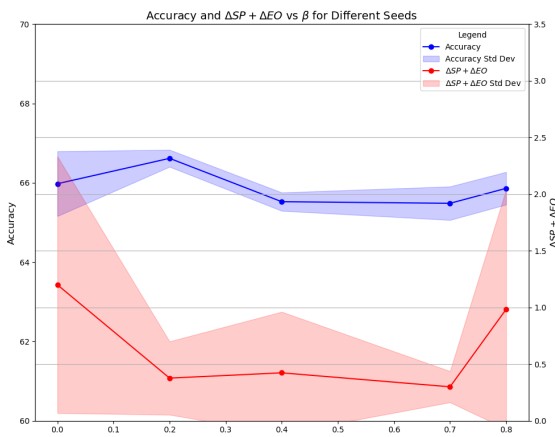

Figure 2: Hyperparameter value study for $\beta$, which influences the trade-off between accuracy and fairness.

## 4.2 Results beyond original paper

In the original study, FairAC was presented as a generic method applicable to a wide range of downstream tasks. This claim was primarily supported by tests on different datasets. Therefore, we performed additional experiments to test this claim. Firstly, our study focused on examining FairAC's performance across various sensitive attributes. Unlike the original experiment that only considered region as the sensitive variable, in

this study age and gender were used as sensitive variables. The findings, detailed in Table 3, show that gender as sensitive attribute produces results comparable to those obtained with region. However, for age, the fairness performance decreases drastically. In this scenario, the standard GCN model outperforms both fair models. This outcome is due to age being an important feature for the final prediction tasks, which challenges the achievement of set accuracy and AUC thresholds. If the thresholds are set lower, the accuracy decreases but the fairness increases. Experiments on different thresholds values can be found in Appendix E.

| Sensitive attribute | Method | Acc ↑ | AUC ↑ | ΔSP ↓ | ΔEO ↓ | ΔSP+ΔEO ↓ | Consistency ↑ |
|---|---|---|---|---|---|---|---|
| Region | GCN | $65.10 \pm 0.24$ | $68.42 \pm 0.12$ | $1.72 \pm 1.17$ | $1.37 \pm 0.51$ | $3.08 \pm 1.68$ | $41.35 \pm 0.01$ |
| | FairGNN | $\mathbf{68.16 \pm 0.59}$ | $\mathbf{75.67 \pm 0.52}$ | $1.56 \pm 0.45$ | $3.17 \pm 1.07$ | $4.73 \pm 1.47$ | $41.35 \pm 0.01$ |
| | FairAC | $65.33 \pm 0.30$ | $71.20 \pm 1.74$ | $\mathbf{0.55 \pm 0.10}$ | $\mathbf{0.13 \pm 0.15}$ | $\mathbf{0.68 \pm 0.09}$ | $41.33 \pm 0.00$ |
| Gender | GCN | $63.40 \pm 0.20$ | $68.56 \pm 0.40$ | $3.28 \pm 0.71$ | $2.97 \pm 0.61$ | $6.24 \pm 1.13$ | $41.35 \pm 0.01$ |
| | FairGNN | $64.25 \pm 0.41$ | $72.25 \pm 2.49$ | $2.27 \pm 0.95$ | $2.63 \pm 0.31$ | $4.90 \pm 0.77$ | $41.35 \pm 0.01$ |
| | FairAC | $\mathbf{66.44 \pm 0.47}$ | $\mathbf{73.39 \pm 0.20}$ | $\mathbf{0.66 \pm 0.56}$ | $\mathbf{0.30 \pm 0.10}$ | $\mathbf{0.96 \pm 0.52}$ | $41.33 \pm 0.00$ |
| Age | GCN | $64.94 \pm 1.11$ | $71.33 \pm 1.94$ | $\mathbf{24.69 \pm 3.21}$ | $20.57 \pm 3.84$ | $\mathbf{45.26 \pm 6.96}$ | $41.35 \pm 0.01$ |
| | FairGNN | $65.79 \pm 0.20$ | $72.53 \pm 1.42$ | $39.83 \pm 4.80$ | $37.23 \pm 2.20$ | $77.07 \pm 6.70$ | $41.35 \pm 0.01$ |
| | FairAC | $\mathbf{65.82 \pm 0.69}$ | $\mathbf{74.26 \pm 0.42}$ | $27.46 \pm 1.94$ | $\mathbf{19.90 \pm 2.52}$ | $47.36 \pm 4.38$ | $41.33 \pm 0.00$ |

Table 3: Comparison of FairAC with FairGNN and GCN on the pokec-z dataset with different sensitive attributes. The methods are applied on the GCN classifier. The best results are denoted in bold.

The FairAC framework's performance was evaluated on two additional datasets beyond those used in the original study, with results detailed in Table 4. In the recidivism dataset, the fairness results are almost perfect across all methods. While FairGNN marginally outperforms FairAC in group fairness, FairAC performs better in individual fairness. The credit dataset shows a similar pattern, with FairAC achieving the best fairness performance among the methods tested, while maintaining comparable overall performance. These outcomes lend further support to the claim that FairAC provides better fairness results than comparable methods and thus support the claim that FairAC is a generic method. Moreover, we extended our analysis by incorporating an additional metric, consistency, as explained in Section 3.4.

Contrary to the common trade-off between group and individual fairness [16], our findings (see Table 1) indicate similar consistency levels across GCN, FairGNN and FairAC. This suggests that improving group fairness does not necessarily compromise individual fairness.

| Dataset | Method | Acc ↑ | AUC ↑ | ΔSP ↓ | ΔEO ↓ | ΔSP+ΔEO ↓ | Consistency ↑ |
|---|---|---|---|---|---|---|---|
| Credit | GCN | $72.16 \pm 4.00$ | $64.80 \pm 0.45$ | $7.43 \pm 0.92$ | $5.97 \pm 0.95$ | $13.40 \pm 1.46$ | $26.46 \pm 0.27$ |
| | FairGNN | $\mathbf{77.19 \pm 0.85}$ | $64.66 \pm 0.46$ | $2.02 \pm 1.84$ | $0.83 \pm 1.10$ | $2.85 \pm 2.91$ | $\mathbf{28.02 \pm 1.41}$ |
| | FairAC | $69.78 \pm 2.94$ | $\mathbf{65.13 \pm 0.07}$ | $\mathbf{0.68 \pm 0.51}$ | $\mathbf{0.50 \pm 0.61}$ | $\mathbf{1.18 \pm 0.29}$ | $27.24 \pm 0.81$ |
| Recidivism | GCN | $62.37 \pm 0.02$ | $63.01 \pm 2.43$ | $0.01 \pm 0.02$ | $0.07 \pm 0.12$ | $0.08 \pm 0.14$ | $\mathbf{3.93 \pm 0.00}$ |
| | FairGNN | $\mathbf{70.00 \pm 0.00}$ | $56.66 \pm 1.46$ | $\mathbf{0.00 \pm 0.00}$ | $\mathbf{0.00 \pm 0.00}$ | $\mathbf{0.00 \pm 0.00}$ | $6.60 \pm 0.00$ |
| | FairAC | $63.03 \pm 1.17$ | $\mathbf{70.32 \pm 13.02}$ | $0.04 \pm 0.08$ | $\mathbf{0.00 \pm 0.00}$ | $0.04 \pm 0.08$ | $\mathbf{3.93 \pm 0.01}$ |

Table 4: Comparison of FairAC with FairGNN and GCN on the pokec-z dataset with different sensitive attributes. The methods are applied on the GCN classifier. The best results are denoted in bold.

## 5 Discussion

In this study, various experiments on reproducing the study Fair Attribute Completion on Graph with Missing Attribute [1] were presented. The results of these experiment support the original claims of the authors. Specifically, FairAC has a better fairness performance than other graph attribute completion methods, while maintaining a comparable accuracy, even when a large part of the attributes is missing. The method uses adversarial learning, which indeed improves the performance. To support the claim that FairAC is a generic method, various additional experiments were performed. From these experiments, it can be concluded that

FairAC is applicable across multiple datasets in different fields. Also, FairAC gives accurate results for various sensitive attributes, while the performance might drop if the attribute is important for the downstream task. To examine the impact of FairAC on individual fairness, an additional metric was implemented, from which can be concluded that there is almost no trade-off between group fairness and individual fairness in this study. Since this is contrary to most findings in literature [16], for future research it would be interesting to look into the causes of a trade-off in individual fairness and group fairness.

The code implementation of this study, together with the refactored code from the FairAC paper, can be found on GitHub[4].

### 5.1 What was easy and what was difficult

The original paper provides a complete codebase of FairAC. This was very helpful, since the codebase also includes the hyperparameters utilized in the experiments of the original paper. This made running the experiments easier. However, understanding the code turned out to be a non-trivial task. A lot of training and implementation details were not mentioned in the paper and the code was not clearly structured or documented. This made it very difficult to adjust the codebase for additional experiments, therefore we refactored all original code. In addition to this, the GCN and FairGNN baselines used in the original paper were not part of the codebase, so these were adapted from the FairGNN codebase and changed to meet the requirements of the FairAC paper.

### 5.2 Communication with original authors

We originally reached out the authors to ask for clarification on the setup on some of their experiments, namely the random seeds used and the exact setup for the BaseAC experiment, since the description of this experiment was inconsistent with the codebase. We received a very quick clear response to these questions. Later in the project, we reached out again to ask some questions regarding ambiguities in the data handling and splitting that was taking place in the code. We received a short response with some clarification regarding the asked questions.

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

# Appendices

## A    Model details

In this section the components of the loss function are described in more detail. As explained in Section 3, the loss function used by FairAC consists of three components: a feature fairness loss ($\mathcal{L}_F$), completion loss ($\mathcal{L}_C$) and topological fairness loss ($\mathcal{L}_T$), with a parameter $\beta$ to control the influence of the sensitive classifier:

$$\mathcal{L} = \mathcal{L}_F + \mathcal{L}_C + \beta \mathcal{L}_T \tag{4}$$

**Feature fairness**. The feature fairness loss consist of two components, an auto-encoder loss and sensitive classifier loss. The feature fairness loss leverages the sensitive classifier $C_s$ to adversarially train the auto-encoder, such that the encoder is able to generate fair feature embeddings that can fool the sensitive classifier.

$$\mathcal{L}_F = \mathcal{L}_{ae} - \beta \mathcal{L}_{C_s} \tag{5}$$

**Auto-Encoder**. The auto-encoder encodes original attributes $\mathcal{X}_i$ to feature embeddings $\mathcal{H}_i$ and reconstructs the attributes $\hat{\mathcal{X}}_i$ from latent space. The reconstructed attributes should be close to the original attributes. The loss optimizes for this:

$$\mathcal{L}_{ae} = \frac{1}{|\mathcal{V}_{keep}|} \sum_{i \in \mathcal{V}_{keep}} \sqrt{\left( \hat{\mathcal{X}}_i - \mathcal{X}_i \right)^2} \tag{6}$$

**Sensitive classifier**. The sensitive classifier takes feature embeddings $\mathcal{H}_i$ as input and predicts the sensitive attribute $\hat{s}_i$. Since the sensitive attribute is binary, binary cross entropy loss is used:

$$\mathcal{L}_{C_s} = -\frac{1}{|\mathcal{V}_{keep}|} \sum_{i \in \mathcal{V}_{keep}} s_i \log \hat{s}_i + (1 - s_i) \log(1 - \hat{s}_i) \tag{7}$$

**Attribute completion**. The attribute completion loss measures the difference between the true feature embeddings and the predicted feature embeddings across nodes whose attributes are intentionally dropped ($\mathcal{V}_{drop}$). This loss is averaged over all such nodes:

$$\mathcal{L}_C = \frac{1}{|\mathcal{V}_{drop}|} \sum_{i \in \mathcal{V}_{drop}} \sqrt{(\hat{\mathcal{H}}_i - \mathcal{H}_i)^2}. \tag{8}$$

**Topological fairness**. The attribute completion may introduce topological unfairness by assuming topological information is similar to attributes. To address this issue, FairAC leverages the sensitive classifier ($C_s$) to help mitigate this unfairness. The loss function aims to learn feature embeddings that can fool the sensitive classifier ($C_s$) to predict uniformly over the sensitive category:

$$\mathcal{L}_T = -\frac{1}{|\mathcal{V}_{drop}|} \sum_{i \in \mathcal{V}_{drop}} s_i \log \hat{s}_i + (1 - s_i) \log(1 - \hat{s}_i). \tag{9}$$

# B  Datasets

| Dataset | NBA | Pokec-z | Pokec-n | Credit | Recidivism |
|---|---|---|---|---|---|
| # of nodes | 403 | 67,797 | 66,569 | 30,000 | 18,876 |
| # of edges | 16,570 | 882,765 | 729,129 | 200,526 | 321,308 |
| Density | 0.20456 | 0.00038 | 0.00032 | 0.00045 | 0.001804 |
| Sensitive attributes | country | region | region | age | race |
| Predicted labels | salary | working field | working field | no default next month | bail vs. no bail |

Table 5: Statistics of five graph datasets, namely nba, pokec-z, pokec-n, german credit and recidivism.

In Table 5, the statistics for all five datasets used in this study are given. There is a big difference in size between all datasets. Pokec-z and Pokec-n are the largest dataset, followed by Credit, Recidivism and NBA, respectively. NBA is a dataset with only a small number of nodes. However, the density of NBA is a lot larger than the density of all other dataset. Lastly, the default sensitive attributes are displayed, together with the node labels that are predicted in the downstream node prediction task.

# C  Model training details

---
**Algorithm 1** Model training

---
**Input:** $\mathcal{G} = (\mathcal{V}, \mathcal{E}, \mathcal{X}), \mathcal{S}$
**Output:** auto-encoder $f_{AE}$, Sensitive classifier $C_s$, Attribute completion $f_{AC}$
 1: Obtain topological embedding $\mathcal{T}$ with DeepWalk
 2: **repeat** Pre-train $f_{AE}$
 3:    Obtain the feature embeddings H with $f_{AE}$
 4:    Optimize $f_{AE}$ to prevent unstable embeddings by $\mathcal{L}_{ae} + \mathcal{L}_C$
 5: **until** 200 iterations
 6: **repeat** Train $f_{AE}$, $C_s$ and $f_{AC}$ using downstream task
 7:    Obtain the feature embeddings H with $f_{AE}$
 8:    Optimize the $C_s$ by $\mathcal{L}_{C_s}$
 9:    Optimize $f_{AE}$ to mitigate feature unfairness by loss $\mathcal{L}_F$
10:    Divide $\mathcal{V}_+$ into $\mathcal{V}_{\text{keep}}$ and $\mathcal{V}_{\text{drop}}$ based on $\alpha$
11:    Obtain the feature embeddings of nodes with missing attributes $\mathcal{V}_{\text{drop}}$ by $f_{AC}$
12:    Optimize $f_{AC}$ to achieve attribute completion by loss $\mathcal{L}_C$
13:    Optimize $f_{AC}$ to mitigate topological unfairness by loss $\mathcal{L}_T$
14: **until** convergence

---

In Algorithm 1, the model training process is shown. First of all, the auto-encoder is pre-trained for 200 epochs. After this, the auto-encoder is optimized, together with the sensitive classifier and the attribute completion mechanism. More details can be found in Section 3.5.1.

# D  Codebase

The source is published at `https://github.com/oxkitsune/fact`. A complete refactor of the original codebase[5] has been completed in order to make the framework significantly easier to apply on various downstream tasks. The published code can be used as a library, with any GNN. The GNN only requires a

---

[5]https://github.com/donglgcn/FairAC

slight modification, as the final layer needs to be excluded in order to train the FairAC model. For more details see the `README.md` file in the repository.

## E Fairness performance on age sensitive attribute with different thresholds

| Threshold | Method | Acc ↑ | AUC ↑ | ΔSP ↓ | ΔEO ↓ | ΔSP+ΔEO ↓ | Consistency ↑ |
|---|---|---|---|---|---|---|---|
| 65% acc threshold 69% auc threshold | GCN | $64.94 \pm 1.11$ | $71.33 \pm 1.94$ | $\mathbf{24.69 \pm 3.21}$ | $20.57 \pm 3.84$ | $\mathbf{45.26 \pm 6.96}$ | $\mathbf{41.35 \pm 0.01}$ |
| | FairGNN | $65.79 \pm 0.20$ | $72.53 \pm 1.42$ | $39.83 \pm 4.80$ | $37.23 \pm 2.20$ | $77.07 \pm 6.70$ | $\mathbf{41.35 \pm 0.01}$ |
| | FairAC | $\mathbf{65.82 \pm 0.69}$ | $\mathbf{74.26 \pm 0.42}$ | $27.46 \pm 1.94$ | $\mathbf{19.90 \pm 2.52}$ | $47.36 \pm 4.38$ | $41.33 \pm 0.00$ |
| 60% acc threshold 64% auc threshold | GCN | $\mathbf{61.85 \pm 2.57}$ | $70.52 \pm 0.80$ | $18.55 \pm 1.91$ | $14.57 \pm 1.25$ | $33.12 \pm 2.29$ | $\mathbf{41.35 \pm 0.01}$ |
| | FairGNN | $59.77 \pm 0.29$ | $69.36 \pm 3.51$ | $19.66 \pm 8.48$ | $17.37 \pm 5.34$ | $37.03 \pm 13.82$ | $\mathbf{41.35 \pm 0.01}$ |
| | FairAC | $61.27 \pm 0.84$ | $\mathbf{73.59 \pm 0.85}$ | $\mathbf{17.14 \pm 2.77}$ | $\mathbf{12.07 \pm 2.44}$ | $\mathbf{29.21 \pm 5.13}$ | $41.33 \pm 0.00$ |
| 50% acc threshold 50% auc threshold | GCN | $\mathbf{53.60 \pm 0.06}$ | $\mathbf{55.34 \pm 1.86}$ | $0.06 \pm 0.06$ | $0.03 \pm 0.06$ | $0.09 \pm 0.08$ | $\mathbf{41.35 \pm 0.01}$ |
| | FairGNN | $53.59 \pm 0.00$ | $54.29 \pm 2.23$ | $\mathbf{0.00 \pm 0.00}$ | $\mathbf{0.00 \pm 0.00}$ | $\mathbf{0.00 \pm 0.00}$ | $\mathbf{41.35 \pm 0.01}$ |
| | FairAC | $53.59 \pm 0.00$ | $54.32 \pm 1.96$ | $\mathbf{0.00 \pm 0.00}$ | $\mathbf{0.00 \pm 0.00}$ | $\mathbf{0.00 \pm 0.00}$ | $41.33 \pm 0.00$ |

Table 6: Comparison of FairAC with FairGNN and GCN on the pokec-z dataset with age as sensitive attribute for different accuracy and auc thresholds. The methods are applied on the GCN classifier. The best results are denoted in bold.

In Table 6, the experiments with different thresholds for the sensitive attribute age are displayed. In the top row, the results of the original threshold are shown. As one can see and as was analysed in Section 4.2, the model embeddings are a lot more unfair than for other sensitive attributes, respectively 47.36 versus 0.96. As the thresholds decrease, the embeddings get more fair, until the point where they reach 0.0. This means that the model is able to produce fair embeddings, but for important attributes, it can cost a lot of performance.

## F GPU usage

| Experiments | #FairAC models | #FairGNN models | #GNN models | #datasets | #seeds | GPU usage (hours) |
|---|---|---|---|---|---|---|
| Main (Table 1) | 1 | 1 | 1 | 3 | 3 | 7.5 |
| Alpha (Table 2) | 8 | 4 | 4 | 1 | 3 | 16 |
| Beta (Figure 2) | 5 | 0 | 0 | 1 | 3 | 7.5 |
| Sensitive attributes (Table 3, Table 6) | 4 | 4 | 4 | 1 | 3 | 10 |
| Other datasets (Table 4) | 1 | 1 | 1 | 2 | 3 | 5 |
| | | | | | Total: | 46 |

Table 7: GPU usage for all experiments published in this report. Every FairAC model takes roughly 30 minutes to train, every GNN and FairGNN model takes about 10 minutes to train.

In Table 7, the specifics of the GPU hours used per experiment are displayed. The alpha experiments cost the most GPU hours, since all models, including FairAC without adversarial training, had to be trained for every alpha.

## G Hyperparameters

For training the models, various hyperparameters were set. All hyperparameters used in this study were adapted from the original study. Per default, all models were trained for 3000 epochs, which includes 200 epochs pretraining of the auto-encoder. An initial learning rate of 0.001 was used. On this learning rate, weight decay of $1 \cdot 10^{-5}$ is applied. While training, a dropout of 0.5 is used. The main auto-encoder has a

hidden dimension of 128 and uses one attention head. The default feature drop rate ($\alpha$) was set to 0.3, unless mentioned otherwise. $\beta$ was set to 1.0 for all datasets, expect for the pokec-n dataset, where it was set to 0.5. As an accuracy threshold, 65.0 was used and for the auc threshold, 69.0 was used. The hyperparameters used for all dataset are shown in Table 8. For the dataset that were not used in the original study, the hyperparameters were adapted from Dong, Ma, Wang, *et al.* [19].

| Dataset | Minimum number of datapoints used for AC training | Minimum number of datapoints used for sensitive classifier training |
|---|---|---|
| Pokec-n | 500 | 200 |
| Pokec-z | 500 | 200 |
| NBA | 100 | 50 |
| Credit | 6000 | 500 |
| Recidivism | 200 | 100 |

Table 8: Hyperparameters specific to the data loading for all datasets used in this study.

In addition to these hyperparameters, the GNN and FairGNN model had to be pretrained. The pretraining was done for 800 epochs. For the pretraining, all other hyperparameters are equal to the normal training hyperparameters.

FairAC uses topological input embeddings which are created using Deepwalk [20]. This embedding was created using 10 epochs with a walk length of 100 and a window size of 5. 4 workers were used, and the dimension adapted was 64. Lastly, a learning rate of 0.05 was adapted.

