# OpenReview forum: "Reproducibility study of FairAC"
_TMLR — Accepted by TMLR_

### Review · Reviewer_ovzk · 2024-03-25

**Summary Of Contributions:**

The paper "Reproducibility study of FairAC" reproduces experiments of a previously published fairness paper. The reproduced paper is focused on fairness for graph attributes completion. After code refactor, It observes same results as original ones on the same datasets, and similar tendencies on others.

**Audience:**

No

**Broader Impact Concerns:**

.

**Claims And Evidence:**

No

**Requested Changes:**

- A large amount of formalization / more detailled presentation / discussion is needed before the paper could be published in TMLR from my point of view. Reading this paper alone is clearly not enough for the reader, it does not allow to understand the differences of the considered approach with its competitors in the field.
- FairGCN is mentioned without any reference nor any explanation about what this method is
- I feel authors should rather focus maybe on an innovation to improve individual fairness in the context of graph conception, to better perform on that task they introduce in the paper, rather than only limiting on observing performances from others

**Strengths And Weaknesses:**

Strengths:
   - Consistent results with the original paper
   - Refactored code released with baselines

Weaknesses:
   - I am not sure that such reproducibility paper is well fitted with journal venues as TMLR. Some workshops are specifically dedicated for this
   - More importantly, I am not sure that the paper brings much beyond what we already knew from the original paper, except that results can be reproduced. New experiments are not so different. In what really differ the new datasets considered ?
   - The paper is really not self-contained and really lacks of formalization of the task / presentation  of the method / discussion of the field
   - Not any discussion about what should be done to go further, beyond some limits of the approach

---

> ### Author Response · Authors · 2024-04-25
>
> Dear reviewer,
>
> Thank you for your review and the comments on our paper. We appreciate the comment about the refactored code leading to results consistent with the original paper, since this was the main focus of our submission. Namely, this submission was created with the intent to become part of the ML Reproducibility Challenge. With this challenge in mind, we created a reproducibility study with extensions, while also improving the overall reproducibility of the original work by improving the code.
>
> Our work goes beyond the original paper by containing an individual fairness analysis of the methods used in the original study. Namely, since in literature, often a tradeoff between individual fairness and group fairness is observed. In our study, this tradeoff is researched with the conclusion that applying FairAC to obtain more group fairness does not come at the cost of individual fairness. Based on your feedback, in our revised version, we highlighted this result more.
>
> We hope this addresses the points you raised, and look forward to your response.

---

### Review · Reviewer_mddg · 2024-03-26

**Summary Of Contributions:**

The paper reproduces FairAC method and conduct some new experiments on additional settings. The reproduction shows the effectiveness of the FairAC method and also reveals additional findings beyond that. Specifically, the author finds out that in some situation, the trade-off between group and individual fairness is marginal.

**Audience:**

Yes

**Claims And Evidence:**

Yes

**Requested Changes:**

1. It is better to investigate the reason behind the experiment results, such as propose new method to evaluate that or based on that to further revise it.
2. The writing could be improved by making the presentation concise and accurate.

**Strengths And Weaknesses:**

Strengths:
1. It studies the trade-off between group fairness and individual fairness.
2. It revise the codebase which accelerate the training process.

Weakness:
1.  It has limited novelty. Other than reproduction, it does not propose new method or improvement.
2. The presentation is redundant. The paper looks like a report rather than a technical paper and includes many details that are not particularly informative.

---

> ### Author Response · Authors · 2024-04-25
>
> Dear reviewer,
>
> Thank you for your review and the comments on our paper. We appreciate the comment about the individual fairness tradeoff study, since this was a large part of our new research. Namely, this submission was created with the intent to become part of the ML Reproducibility Challenge. With this challenge in mind, we created a reproducibility study with extensions, while also improving the overall reproducibility of the original work by improving the code. This also explains the many technical details in the submission such as the specific GPU used for training, alongside its power consumption. These were included to make the research as transparent as possible.
>
> To include the feedback, we have made parts of our submission more concise while not losing important details for reproducibility.
>
> We hope this addresses the points you raised, and look forward to your response.

---

### Review · Reviewer_Hp7d · 2024-04-12

**Summary Of Contributions:**

This submission is a reproducibility report of an existing paper FairAC: "Fair Attribute Completion on Graph with Missing Attributes". The submission reports the main technical contribution of that paper, reproduces the main empirical results, and extends the study to more datasets, more sensitive attributes, and more fairness criteria. The submission verifies that the results of the original paper are reproducible and the claims hold.

**Audience:**

No

**Broader Impact Concerns:**

There is no major broader impact concern. Adding a statement or discussion about the fairness definitions and their limitations would be great.

**Claims And Evidence:**

Yes

**Requested Changes:**

There are no specific requested changes from my side. However, I think adding more discussion and connecting the work to more existing literature would further improve the submission quality. Moreover, I would recommend the work submitting to the ML Reproducibility Challenge at https://reproml.org/.

**Strengths And Weaknesses:**

Strengths:
- The submission is a comprehensive reproducibility study, covering various aspects of the result reproduction, including both the theoretical front and engineering front. For practitioners, the submission would be of high value.

- The writing quality is high and it is easy to follow.

Weaknesses:
- As a reproducibility study, the submission may be adding limited value to the existing literature. A large portion of the submission verifies the existing claims and research findings instead of proposing or presenting novel things. Though the study extends the original empirical results to more dataset, more sensitive attributes, and more fairness criteria, the novel findings and discussions are relatively few (mostly in Section 4.2).

---

> ### Author Response · Authors · 2024-04-25
>
> Dear reviewer,
>
> Thank you for your review and the comments on our paper. This submission was indeed created with the intent to become part of the ML Reproducibility Challenge. With this challenge in mind, we created a reproducibility study with extensions, while also improving the overall reproducibility of the original work by improving the code.
>
> To include feedback, we have improved the definition for individual fairness, and look forward to your response.

---

### Decision · Action_Editor_yKa2 · 2024-05-31

**Recommendation:** Accept as is

**Comment:**

This paper is a reproducibility study that verifies whether the experiments in the FairAC paper is reproducible. The submission conducts a comprehensive reproducibility study, covering various aspects of the result reproduction, including both the theoretical front and engineering front. Also, the experiments were done on more datasets than in the original paper, showing the generalizability and robustness of the original method. The results presented in the paper are useful for practitioners. Though the reviewers are negative about this paper due to lack of novelty, I believe it meets the acceptance criteria for reproducibility papers in TMLR. I would recommend acceptance of this submission.

**Audience:**

yes

**Claims And Evidence:**

yes